# Assessment of the Corrosion Resistance of Thermal Barrier Coatings on Internal Combustion Engine Components

**DOI:** 10.3390/ma18061227

**Published:** 2025-03-10

**Authors:** Daniela Lucia Chicet, Jozsef Juhasz, Cosmin Mihai Cotruț, Bogdan Istrate, Corneliu Munteanu

**Affiliations:** 1Faculty of Materials Science and Engineering, Gheorghe Asachi Technical University of Iasi, Blvd. Mangeron, No. 41, 700050 Iasi, Romania; daniela-lucia.chicet@academic.tuiasi.ro; 2Faculty of Engineering, Technical University of Cluj-Napoca, North University Center of Baia Mare, V. Babes St. 62A, 430083 Baia Mare, Romania; 3Faculty of Materials Science and Engineering, National University of Science and Technology Politehnica Bucharest, 313 Independentei Street, 060042 Bucharest, Romania; cosmin.cotrut@upb.ro; 4Faculty of Mechanical Engineering, Gheorghe Asachi Technical University of Iasi, Blvd. Mangeron, No. 61, 700050 Iasi, Romania; bogdan.istrate@academic.tuiasi.ro; 5Technical Sciences Academy of Romania, 26 Dacia Blvd., 030167 Bucharest, Romania

**Keywords:** thermal barrier coatings, atmospheric plasma spray, corrosion resistance

## Abstract

Thermal barrier coatings (TBCs) can be applied on the inner surface of the combustion chamber of internal combustion engines to reduce fuel consumption and pollution and also improve the fatigue life of their components. The purpose of the present work was to evaluate the corrosion resistance in an environment equivalent to the one generated by combustion gases for three types of TBCs—P1 from Cr_3_C2-25(Ni20Cr), P2 from MgZrO_3_-35NiCr and P3 from ZrO_2_-5CaO—with all of them having a base coat from Al_2_O_3_-30(Ni20Al) powder. The coatings were deposited via atmospheric plasma spray (APS) on the intake/exhaust valves of a gasoline internal combustion engine, both before and after their use in operation (Dacia 1400 model, gasoline fuel, Dacia Company, Mioveni, Romania). The samples were studied from the electrochemical corrosion resistance point of view, and their morphology and structure were analyzed using SEM, EDS and XRD methods. After analyzing the results of the samples before and after testing them in operation, it was observed that the presence of the coatings improved the corrosion resistance of the material used for the production of the valves.

## 1. Introduction

In the automotive industry, customer demands need to be aligned with environmental requirements, so manufacturers must constantly find new ways to improve functionality, robustness, comfort, safety and environmental protection [1]. As is well known, the need to limit greenhouse gas emissions (GHG emissions) is an imperative one, recognized both at the European level in the framework of “The European green deal” Package [2] and at the global level during the last UN conference: COP29 (Baku, Azerbaijan, Nov. 2024). Here, leaders recognized the need for global efforts to maintain, within the realm of the possible, the objective of limiting global warming to 1.5 °C by taking immediate action to reduce global GHG emissions by 40% by 2050 [3]. According to the Kyoto Protocol [4], GHGs are defined as non-fluorinated and fluorinated gases, of which methane, nitrous oxide and carbon dioxide have a major impact on global warming (about 90% of the greenhouse effect), with the latter 24% of the global total being produced by the transportation industry through the use of fossil fuels [5].

In response to these requirements, research in the field of reducing the impact of the use of fossil fuels has developed in three major directions: the use of fully electric vehicles, fuel efficiency [6] and the continuous optimization of internal combustion engines (ICEs) to reduce consumption [7]. This third goal can be achieved through several constructive approaches, one of them being combustion chamber insulation, which brings with it several advantages: increasing the turbocharging efficiency by increasing the temperature of the exhaust gases; increasing the lifetime of components by decreasing the thermal stress they are subjected to and reducing the costs associated with cooling systems by decreasing the thermal energy absorbed by the cooling systems [8,9,10].

Although by no means new, one of the current methods of insulation is the coating of engine components, which are subject to extreme solicitations, via thermal deposition technology using layers resistant to wear, corrosion (oxidation), erosion, high temperatures and thermal shock [11,12,13,14,15,16], generically called TBCs (Thermal Barrier Coatings). The studies on their fabrication and performance started about four decades ago [17,18], with great success in the aerospace industry, where they have been used to coat gas turbine components [19,20] and aircraft engine components [21,22]. For this reason, in recent years, the use of TBCs for heat transfer process control has been extended to other fields, one of them being the automotive industry, where it is mainly used for coating the main components of internal combustion engines [23,24,25,26]. A modern variant, generically called a Low-Heat-Rejection (LHR) engine, has been obtained by applying a thermal barrier coating on its main components, including the cylinder block, cylinder head, piston surface and valves [27,28], which acts as a thermal insulating layer, minimizing heat losses. Thus, the thermal energy lost in conventional engines (e.g., about one-third in diesel engines) can be reused to increase the fuel conversion efficiency [29], leading to increased engine power and thermal braking efficiency and reduced hydrocarbon and smoke emissions [30]. Another benefit of using TBCs is the extended service life of the components exposed to thermal fatigue [31], as they are designed to sustain thermal duty cycles inside the engine cylinder between 700 and 800 °C.

A TBC-type coating system is mainly composed of three parts [32]: (1) the substrate, which is usually a high-temperature alloy but can also be a common alloy; (2) a bond coat to the substrate, called BC, and (3) a top coat that comes in direct contact with the working environment, called TC, each with distinct physical, chemical and thermo-mechanical properties. Due to this, the risk of failure of TBCs in severe operating environments increases dramatically, as the differences between the properties of the component layers are greater. A second cause of failure via exfoliation of TBCs is the formation and growth of a transition oxide (TGO—Thermally Grown Oxide) layer [33]. Delamination of the surface layer of TBCs usually occurs along the spinel-type TGO, and the generation of the TGO induces additional stresses, causing the exfoliation of the top layer due to interconnecting cracks [7].

For these reasons, many combinations of the consecrated materials have been studied to ensure the best efficiency of TBC-type systems applied to internal combustion engines, using a wide variety of thermal spray methods: APS (atmospheric plasma spray), HVAF (High-Velocity Air Fuel), SPS (Suspension Plasma Spray), VPS (Vacuum Plasma Spray), HVOF (High-Velocity Oxygen Fuel) a.s.o. The most recent research is summarized in Table 1.

As observed from the literature survey, applying TBCs enhances thermal efficiency by minimizing heat losses through the combustion chamber walls. This efficiency gain ensures more complete combustion of the fuel–air mixture, directly reducing CO and HC emissions. Additionally, their ability to elevate exhaust temperatures can improve the effectiveness of after-treatment systems, further lowering these emissions. Paparao et al. [27] demonstrated that YSZ-CeO_2_ coated pistons reduced CO emissions by 44.1% and HC emissions by 46.7% in dual-fuel engines. This reduction was attributed to the improved thermal efficiency provided by the TBC layer, which minimizes heat loss and enhances the oxidation of CO and HC during combustion. Moreover, the study highlighted that the combined use of TBCs and hydrogen-enriched fuels further lowered emissions due to the better combustion characteristics of hydrogen. Zheng et al. [8] found that MgZrO_3_-coated pistons in gasoline compression ignition (GCI) systems reduced HC and CO emissions, especially under low-load conditions. The coatings improved combustion stability by increasing in-cylinder temperatures, critical for achieving lower emissions in GCI engines, where fuel evaporation and mixing are key challenges. Liu et al. [22] conducted simulations showing that a ceramic layer of 370 µm applied to the piston’s surface reduced throat temperatures by over 50 °C, contributing to significant CO and HC emission reductions. These findings align with those of Gautam et al. [52], who reported that plasma-sprayed zirconia coatings reduced fuel consumption by 10% and HC and CO emissions by up to 40%, albeit with a 7–11% increase in NOx emissions.

Regarding the influence of TBC material properties on emissions, it has been observed that coatings with low thermal conductivity increase the in-cylinder temperatures, thereby intensifying NOx formation [53]. Nevertheless, optimized porosity can simultaneously reduce heat losses and NOx emissions [38]. Other significant properties include surface roughness and coating thickness, as analyzed by Gingrich et al. [17]. They demonstrated that thicker coatings with smoother surfaces improve thermal efficiency and reduce CO and HC emissions, although they may increase NOx emissions. Advanced materials have also been proposed, such as lanthanum zirconate (La_2_Zr_2_O_7_), as suggested by Karthikayan et al. [54]. This material exhibited superior resistance to thermal cycling and lower thermal conductivity, contributing to reduced CO and HC emissions without amplifying NOx emissions. Numerical models developed by Yao et al. [37] and experimental validations by Somhorst et al. [5] highlight the potential of advanced materials such as PSZ and La_2_Zr_2_O_7_ to optimize emissions.

When analyzing the operating conditions and TBC performance, it was observed that under high-load conditions, the effects of TBCs on emissions are limited, as the already high temperatures exacerbate NOx formation [8,17]. Under low-load conditions, TBCs improve combustion stability and reduce emissions [32]. Multi-layered coatings and optimized thicknesses are critical for balancing thermal efficiency and NOx emissions [37].

Starting from the observation that, in the literature, the attention has been mainly focused on the study of the effect of the coatings acting as a thermal barrier on the cylinder head and piston surface of diesel engines, we considered it opportune to evaluate the effect of such a type of coating on other elements that compose the combustion chamber, namely the intake and exhaust valve plates. Another aspect that is not very frequently encountered in the literature when evaluating the operating conditions of TBCs deposited on the inside walls of the combustion chamber is the corrosive effect of the combustion gases, which can substantially affect their performance by affecting the cohesion with the substrate. Thus, the present study aims to evaluate corrosion resistance in an environment equivalent to that generated by combustion gases for three types of TB coatings deposited by atmospheric plasma spray on the intake/exhaust valves of a gasoline internal combustion engine, both before and after their use in operation.

## 2. Materials and Methods

### 2.1. Sample Material

In this experiment, four sets of commercially available valves (intake and exhaust) were considered, one of the sets being kept as the control (R). The base material for the intake valves was Cr-Si steel, and the one for the exhaust valve was austenitic stainless steel.

The three other sets were coated with three TBC-type layers, as follows:–The bond coat (BC) was made from Al_2_O_3_-30 (Ni20Al) powder, produced by Metco-Oerlikon under the name 410NS.–The top coat for set 1 (P1) was made from Cr_3_C_2_–25(Ni20Cr) powder, produced by Metco-Oerlikon under the name 81NS.–The top coat for set 2 (P2) was made from MgZrO_3_-35NiCr powder, produced by Metco-Oerlikon under the name 303NS.–The top coat for set 3 (P3) was made from ZrO_2_-5CaO-0.5Al_2_O_3_-0.4SiO_2_ powder, produced by Metco-Oerlikon under the name 201NS.

The bond coat was chosen from the metallic–ceramic blend powder type, recommended for thermal barrier or thick clearance control applications, where thermal expansion mismatch between the substrate and the coating must be very well controlled. The metallic component is a chemically clad nickel aluminum (Ni 20%Al), and the ceramic one is a fused and crushed gray aluminum oxide. This combination ensures a coating that is very hard, smooth, denser, stronger and more resistant to abrasion and shock than coatings of pure ceramic, concurrent with a higher coefficient of thermal expansion, as well as less susceptible to cracking than pure ceramic. As presented in the specialized literature [55], the presence of a Thermally Grown Oxide (TGO) layer with a thickness of 2–3 microns is sufficient to protect the substrate from the effects of thermal shock. Furthermore, it has been demonstrated that uncontrolled oxidation of the bond coat (BC), along with TGO growth, can generate excessive internal stresses, leading to a reduction in interlayer adhesion, which may ultimately result in delamination or spallation [56]. Based on these observations, we have chosen a BC formulation that already contains alumina (Al_2_O_3_) as a thermal barrier component while being coupled with a metallic matrix to ensure substrate adhesion. The selection of this powder was in accordance with the manufacturer’s recommendations [57].

For the top coat, a representative powder was selected from each of the three major categories of powders used in thermal spray processes: metallic powders (P1), cermet powders (P2) and ceramic powders (P3). The selection was based on references from the specialized literature (see Table 1) and the manufacturer’s specifications [58,59,60]. The powders were chosen to have, as the main criterion, high-temperature wear and corrosion resistance (higher than 800 °C), together with other surface functions, such as abrasion protection, corrosion resistance (in corrosive gas/liquid), erosion protection in gas flow and high-temperature oxidation resistance.

### 2.2. Coating Deposition

To obtain the samples using plasma jet deposition in a normal atmosphere (method usually known as APS) with the Spraywizard 9MCE-type system (Metco-Oerlikon, Singapore, 2008), the following steps were taken:(a)Three sets of valves were established, and valve stems were protected with adhesive metallic paper in order to not be affected during the thermal spray deposition process.(b)The valve plates were sandblasted for surface texturing and cleaned with isopropyl alcohol.(c)The three sets of valves were mounted separately on the turning table of the spraying system with the help of specially made holders (Figure 1a).(d)BCs were deposited on all the valve plates simultaneously, followed by the three types of top coats, respecting the spray parameters indicated by the manufacturer for each type of powder. The as-coated aspect of the samples is presented in Figure 1b.

The spraying parameters used to obtain the coatings were as follows: the N_2_ and H_2_ flow was 3.,4 Barr (the same for all powders); the Ar (carrier gas) flow was 5.66 NLPM for the BC and P1, respectively, and 5.5 NLPM for P2 and P3; the voltage was 70 ÷ 80 V for the BC, respectively, and 70 ÷ 80 V for P1, P2 and P3; the intensity was 500 A for all powders; the spraying distance was 153 mm for the BC, 90 mm for P1, 120 mm for P2 and 110 mm for P3. The feed rate was also different for each powder: 68 g/min for the BC, 91 g/min for P1, 65 g/min for P2 and 87 g/min for P3.

After the deposition was finished, one sample from each batch was metallographically prepared for cross-section analysis, and it was concluded that the thickness of the bond coat had a medium value of 170 µm, and that of the top coats was 75 µm.

### 2.3. “In Situ” Testing

To evaluate the behavior “in operation” of the studied coatings, the four valve sets were mounted, in order, on a test stand realized from an adapted Dacia 1400 model engine (Dacia Company, Mioveni, Romania), using gasoline as fuel, and tested for 36 h at an alternate speed regime, as presented in a previous paper [61]. The samples were used in their as-coated state, without further mechanical surface treatment to prevent contamination.

### 2.4. Corrosion Tests

Corrosion resistance tests were performed using a Potentiostat/Galvanostat (PARSTAT 4000 type, manufactured by Princeton Applied Research, Oak Ridge, TN, USA), to which a low-current interface (VersaSTAT LC, manufactured by Princeton Applied Research) was coupled. The electrochemical tests were performed according to ASTM G5–94 (2011) [62].

After completing the tests on the engine stand, from each set of worn valves, a sample was taken to evaluate the corrosion behavior in an environment equivalent to that generated by the combustion gases. Those samples were codified by adding a “worn” description, as presented in Table 2. To achieve a complete understanding of the corrosion behavior, samples of the coatings in the initial state were also taken and tested.

A standard corrosion cell composed of a saturated calomel electrode (SCE—reference electrode), a platinum electrode (counter electrode) and the working electrode, which consisted of the experimental samples to be investigated, was used for corrosion behavior experiments. Prior to the corrosion tests, the samples were prepared in order to ensure electrical contact followed by the isolation of the electrical contact and of the entire sample, leaving only the surface of interest to be exposed to the electrolyte. Thus, to ensure electrical contact, an insulated Cu wire was micro-welded on the surface opposite the thermal sprayed surfaces. The electrical contact and the entire sample (except the surface of interest) were then covered with silicone (Figure 2). Subsequently, using macroscopic images of each embedded sample and ImageJ software, the area of the non-silicone-coated surfaces was measured and then used for the calculation of the electrochemical parameters. The electrochemical cell was inserted into a Faraday cage to eliminate interference from electromagnetic fields during the corrosion tests.

The corrosion resistance was determined using the linear polarization technique, which consists of plotting linear polarization curves involving the following steps:Measuring/monitoring the open-circuit potential (OCP) over 3 h;Linear polarization resistance (LPR) from −30 mV (vs. OCP) to +30 mV (vs. OCP), with a scanning rate of 0.167 mV/s;Marking the linear polarization curves from −200 mV (vs OCP) to +200 mV (vs. OCP) and Tafel curves, with a scanning rate of 0.167 mV/s.

The corrosion tests were conducted at 25 ± 1 °C, using, as the electrolyte, a solution whose composition was equivalent to the condensate of the exhaust gases produced by a gasoline engine [63], hereinafter referred to as the equivalent solution. The chemical composition of the equivalent solution is as follows: [Cl^−^] = 160 ppm, [SO4^2−^] = 500 ppm, [CO^3−^] = 300 ppm and [NO^3−^] = 10 ppm, with a pH value of 3.

The tested samples were encoded for the corrosion tests as presented in Table 2.

## 3. Results and Discussions

### 3.1. Corrosion Behavior

For a good comparison of the results, they were superimposed in a graph for both the evolution of the OCP (Figure 3) and the Tafel curves (Figure 4).

Following the corrosion tests carried out in the equivalent solution, some parameters were determined to characterize the corrosion resistance of the investigated samples:Open-circuit potential (E_oc_);Corrosion potential (E_corr_);Corrosion current density (i_corr_);The slope of the cathode curve (β_c_);The slope of the anodic curve (β_a_).

With the help of the values determined from the Tafel curves, the following parameters were also calculated to characterize the corrosion resistance of the investigated samples:Polarization resistance (R_p_);Coating porosity (P);Efficiency during a corrosive attack (P_e_).

The polarization resistance (R_p_) was calculated according to ASTM G59-97 (2014) [64] using Equation (1):(1)$$R_{p}=\frac{1}{2.3}\frac{\beta_{a}|\beta_{c}|}{\beta_{a}+\left|\beta_{c}\right|}\frac{1}{i_{\mathrm{cor}}}$$
where β_a_—the slope of the anodic curve; 

β_c_—the slope of the cathode curve.

For the initial state, the coating’s porosity (P) and the efficiency during a corrosive attack (P_e_) were calculated using Equations (2) and (3) [65,66]:(2)$$P=\left(\frac{R_{\mathrm{ps}}}{R_{p}}\right)\times 10^{-\frac{\left|\Delta E_{i=0}\right|}{b_{a}}}$$
where R_ps_—substrate polarization resistance;

R_p_—the coating’s polarization resistance;

ΔE_i=0_—the difference between the values of the corrosion potentials of the substrate and coating;(3)$$P_{e}=\left(1-\frac{i_{\mathrm{corr},\mathrm{coat}}}{i_{\mathrm{corr},\mathrm{substrate}}}\right)\cdot 100$$

i_corr,coat_—coated specimen corrosion current density;

i_corr,substrate_—substrate specimen corrosion current density.

Table 3 presents the main parameters of the electrochemical corrosion process, resulting from the tests performed in the equivalent solution.

The results were evaluated from the point of view of the anticorrosive protection offered by the coatings, comparing the results on samples in the initial state with the results of the worn samples.

#### 3.1.1. Evaluation of Corrosion Resistance Before Performing the Functional Tests

If we take into account the electrochemical measurements presented in Table 3, from the open-circuit potential (E_oc_) evolution point of view, we observe that sample P2i has a more electropositive potential (−463 mV) and, consequently, a more noble electrochemical character and, possibly, a better corrosion behavior. The same sample (P2i) also recorded a more electropositive corrosion potential (E_corr_) (−442 mV), i.e., a better corrosion behavior in the equivalent solution compared to the other three samples.

Another parameter used to evaluate the corrosion behavior is the corrosion current density (i_corr_), a small value indicating good corrosion resistance. In this case, the P1i sample recorded the lowest value (8.751 µA/cm^2^), demonstrating a better corrosion resistance compared to the support material but also to the other types of coatings. This behavior of sample P1i was also highlighted by the recording of polarization resistance (Rp), which had the highest value (8.927 kΩ × cm^2^).

Considering the porosity values of the coatings (P) calculated based on the resulting electrochemical parameters, sample P1i, with a value of only 1.99%, is highlighted. Taking into account the efficiency of the corrosive attack (P_e_), it is observed that the entire sample P1i is noted, with a value of 99.2%.

Comparing the values of the electrochemical parameters resulting in the case of the coated samples with those obtained in the case of the uncoated material (substrate), it can be said that the first demonstrates a better corrosion resistance than the substrate. Of these, the P2i sample is highlighted by a more electropositive potential and by a more electropositive value of the corrosion potential (E_corr_), while the P1i sample is characterized by the lowest corrosion current, the highest resistance to polarization, the smallest porosity and the most efficient corrosive attack resistance.

#### 3.1.2. Evaluation of Corrosion Resistance After Performing the Functional Tests

After performing the functional tests on the stand, from the measurements presented in Table 3, it was observed that the P3u sample recorded parameter values that indicate the best corrosion behavior in relation to the other samples in the worn state: the noblest character from the electrochemical point of view (E_oc_ = 448 mV), the most electropositive corrosion potential (E_corr_ = 511 mV), the lowest corrosion current density (i_corr_ = 2.717 µA/cm^2^) and the highest polarization resistance value (R_p_ = 33.554 kΩ × cm^2^).

When evaluating the overall effect of coatings on the corrosion resistance in the case of the samples tested under real conditions, it was observed that all layers (P1u, P2u and P3u) show a better corrosion behavior in the electrolyte solution compared to the base material (Ru).

### 3.2. Surface Morphology of the Samples Subjected to the Corrosion Tests

In order to supplement the results obtained from the corrosion tests, observations were made of the morphology of the sample surfaces using scanning electron microscopy. The analyses were performed using a Quanta 200 3D Dual-Beam electron microscope (FEI, Eindhoven, The Netherlands, 2008) equipped with the chemical analysis module produced by EDAX-Ametek. The Low-Vacuum mode and Large-Field Detector (LFD) were used, and the secondary electron (SE) images presented in Figure 5, Figure 6, Figure 7, Figure 8, Figure 9, Figure 10, Figure 11 and Figure 12 were acquired at magnification powers between 200× and 8000×.

Figure 5a shows the morphology of the P1i sample surface, which has a specific appearance of the surfaces produced by the atmospheric plasma spray: a porous appearance due to the “in-stack” deposition of the powder particles accelerated toward the substrate in the molten and semi-molten states, with the presence of micro-cracks produced at the moment of solidification at high speed. Figure 5b shows a representative area of the coating’s surface, which was analyzed from the chemical composition point of view (see Table 4). It was observed that the area marked with number 2 has a higher concentration of Cr and less of Ni compared to area 3, being the result of the deposition of some particles with a high Cr concentration.

When analyzing the surface of sample P1u, we observed a major change in its appearance with the formation of a film, as shown in Figure 6a. The elemental chemical analysis carried out on area 1 showed that it is rich in carbon, as presented in the EDS spectra emitted (Figure 6c) and in Table 4. Figure 6b shows the details of area 1, being analyzed also from the chemical composition point of view of two areas with distinct appearances; area 2 is characterized by a higher concentration of P, S, Ca, Zn and Mg and smaller amount of O compared to area 3, as presented in Table 4.

Based on several studies in the literature [67,68,69,70,71] that analyze the formation of residual layers (“tribofilm”) after the exposure of internal combustion engine components to real operating conditions, we concluded that the same phenomenon occurred in our study. The formation of these films was studied in detail by R. Flo et al. [65], who have shown that they are due to the deposition of combustion residues on the surfaces with which the combustion gases come into contact.

In Figure 7a, the surface of sample 2 in the initial state is analyzed, showing, as in the case of sample P1i, the same specific structure of the coatings made via thermal spraying but with a lower porosity than the one of sample 1 and with fewer artifacts. From the chemical composition point of view, as presented in Table 5, the presence of O is confirmed, which attests to the formation of oxidized layers during the spraying process. The different percentages of Cr, Ni and Zr observed at the analysis of points 2 and 3 of sample P2i (see Figure 7b,c) indicate the existence of splats with different chemical compositions, with the results depending on the arrangement of the sprayed particles in the molten/semi-melted state on the substrate surface.

Figure 8a,b present the morphology of the P2u sample surface, where a residue film with a frail appearance is visible. According to the R. Flo et al. studies regarding the “tribofilm” dynamics [64,65], we could consider that the residue film is in its equilibrium phase, almost fully covering the surface. Its existence is also confirmed by the chemical analysis (see Table 5), which attests to a high weight percent of the carbon accompanied by other specific elements of the TBC, such as Ni, Fe, Zr and Cr.

Figure 9a,b show the surface morphology of sample 3 in the initial state. The same specific aspect is observed for the layers realized via thermal spraying, namely the existence of superficial splats, porosity and superficial micro-cracks caused by the very rapid cooling of the molten particles in contact with the substrate surface.

Figure 10a,b show the presence of the residue film produced after the accumulation of combustion residues, deposited on the surface of the valve plate P3u. The film is interrupted by cracks, with an aspect characteristic of the breakdown stage of the “tribofilm” dynamics according to R. Flo et al. [66,67].

By comparing the elemental chemical analysis of the P3 surface after use with the initial one, as presented in Table 6, we highlight the presence of the following:Carbon: at a large weight percent;Oxygen: specific to the products resulting from the oxidation of the chemical elements of the coatings, respectively, to the oxides resulting from their exposure to the high temperatures of the combustion chamber;Other elements specific to the combustion residues: Na, Mg, Zn and Ca.

In the case of the reference sample (R), whose appearance after the initial testing is shown in Figure 11, we observe very slight roughness of the surface in Figure 11a, respectively, a microstructure specific to the chromium alloy steels used as a substrate, with the granular aspect being given by the chromium carbides, evenly distributed throughout the mass of the material.

Similar to the other three samples, the surface morphology of the reference sample (Ru) after use in the engine operation, shown in Figure 12a, was analyzed. It was observed that in this case, the residue film is in the form of flakes that cover the entire surface of the valve plate, which is specific to the initial stages of film formation according to R. Flo et al. [64,65]. This aspect is highlighted in Figure 12b, where a detailed view of the residue film flake is presented.

The elemental chemical analysis conducted on the sample’s surface in the initial state (Ri) confirmed the chemical composition of the substrate material, as presented in Table 7. The results obtained for the surface analysis after its use in operation (Ru) confirmed the presence of the carbon element, as well as oxygen, zinc and sulfur as components of the residue film.

The cross-sections of samples extracted from the worn valve plates were metallographically prepared and analyzed via the SEM method with an FEG electron microscope (Quattro C, Thermo Fisher, Holland, MI, USA, 2024), using the High-Vacuum mode, an ETD detector, 10 kV electron beam acceleration and a 10 mm working distance, as presented in Figure 13a–c. With the help of the EDS-incorporated system, the main chemical elements from the cross-sections were emphasized on the chemical distribution maps, which were superimposed on the secondary electrons images. As a result, in Figure 13a, the distribution of the main chemical elements that form the substrate (Fe—pink), the bond coat (Al—yellow; Ni—red) and the top coat (Cr—blue; Ni—red) in the case of the P1u sample are visible. The superior surface is colored in light blue, which represents the carbon, with this also being a proof of residue film presence.

In the same manner, we analyzed the cross-section of sample P2u and observed the presence of Fe (green) in the substrate, Al (blue) and Ni (orange) in the bond coat and the presence of Mg (purple), Zr (pink), Cr (yellow) and Ni (orange) in the top coat. The superior side of the sample was also colored light blue, which represents the presence of carbon. Figure 13c represents the cross-section of sample P3u, with Fe (red) in the substrate; Al (green) and Ni (pink) in the bond coat; Zr (pink), Ca (orange) and Si (blue) in the top coat and carbon on the surface of the coating (light blue). Oxygen was not marked on the distribution map because its presence could introduce interpretation errors. In each of the three cases, it can be observed that the TBC layers maintained their integrity throughout the entire cross-section, and the adhesion to the substrate was not negatively influenced by the corrosive environment or the high temperatures to which it was exposed during the 36 h of continuous engine operation.

The observations derived via SEM and EDS analysis were confirmed using X-ray diffraction analysis conducted on the XPERT PRO MD (Panalitycal, Almelo, The Netherlands, 2009) diffractometer in order to evaluate the structural stability of the thermal barrier coatings applied on the valve trays. The analyses were performed comparatively, both on the as-coated coatings (P1, P2 and P3) and after they had been subjected to corrosion tests in the initial state (P1i, P2i and P3i) and in the worn state (P1u, P2u and P3u), with the diffractograms obtained being comparatively presented in Figure 14.

In the case of the first sample (P1), which was obtained from an Al_2_O_3_-30 (Ni20Al) base coat and Cr_3_C_2_-Ni20Cr top coat, it can be observed that in the as-coated state, the main compounds are those specific to the powders used for coating—Al oxide (Al_2_O_3_; COD: 9009682; COD—the identification number from the Crystallography Open Database), elemental nickel (COD: 2100653) and chromium carbide (COD: 9009907; 5910109)—to which iron oxide (COD: 5910083) and chromium oxide (COD: 9016610) are added, resulting from the exposure of the powder to the very high temperatures specific to the APS deposition process. In the wrought state of the initial coating (P1i), nickel oxidation is observed, with the formation of NiCrO_4_ complex oxide (COD: 2009227) and NiO_2_ (COD: 9012317), the main peaks being those of Ni (COD: 9008510) and chromium carbides (COD: 9009907; 5910109). This aspect is no longer valid in the corroded state of the used coating (P1u), where the presence of elemental Ni is decreased due to its oxidation, and the percentage of chromium oxide increases, concomitant with the appearance of carbon (C4 hexagonal, COD: 1101023; C36 monoclinic, COD: 2103217) as the main element composing the residue film, resulting from exposure to the gases generated by the combustion of fuel.

In the case of the second sample (P2), which is composed of the base layer Al_2_O_3_-30 (Ni20Al) and the top layer MgZrO_3_-35NiCr, the presence of elemental nickel is observed, as well as the presence of oxides in the powder (Al_2_O_3_ and MgO, COD: 9006791; ZrO_2_, COD: 9,009,052; complex oxides of Zr and Mg, COD: 9015025) or generated after the coating process (chromium oxide and nickel (II) oxide, COD: 4320500; complex oxide of Ni and Cr). After exposure of the coating to the corrosive environment (P2i), a decrease in the presence of nickel in the elemental state was recorded due to an increase in the percentage of nickel oxide, concomitant with the formation of a new compound: magnesium peroxide (MgO_2_ COD: 9002354). After exposing the layer to the high temperatures generated during combustion (P2u) and to the corrosive environment specific to this type of fuel, it is observed that the complex oxides of Zr and Mg are no longer present on the analyzed surface, only the specific oxides of the two chemical elements being present. Also, similar to the previous case of sample P1u, the predominant element is the one specific to the formed residue film—carbon—observed both in the form of C (C4 hexagonal; C36 monoclinic) and in the form of graphite (noted G, COD:1011061).

The third type of coating investigated is composed of the same type of BC—Al_2_O_3_-30 (Ni20Al)—with a TC of calcium-stabilized zirconium oxide (ZrO_2_-5CaO). On the surface of the as-coated (P3) sample, the presence of the specific phases of the powder used to obtain this TBC is observed: calcium (COD: 9006716) and zirconium oxides, aluminum oxide present in the BC layer and the nickel element. After exposure to the corrosive environment (P3i), additional oxidation of nickel is observed, with the formation of NiO_2_ and NiO_3_, but also of calcium, with the formation of calcium peroxide (CaO_2_, COD: 9006836), the major compound being zirconia, as expected. The presence of iron (II) oxide-FeO_2_ (COD: 9015157) is also noticed, a sign that the porosity of the coating allowed for the oxidation of the substrate. When analyzing the coating after its use in operation, it can be observed that the main phases are carbon (C16 monoclinic, COD: 9012233; C80 orthorhombic, COD: 9005088), forming the residue film generated after fuel combustion, zirconium oxide and calcium peroxide and oxide. Besides these, the presence of Ni oxides, aluminum oxide (corundum) and iron (II; III) oxide of the Fe_3_O_4_ type (COD: 9006248, as a result of exposure to high temperatures during engine operation) is also observed.

## 4. Conclusions

The study highlighted the positive impact of thermal barrier coatings (TBCs) applied via atmospheric plasma spray (APS) on the corrosion resistance of intake and exhaust valves used in internal combustion engines. The comparative analysis of the samples, both in their initial state and after exposure to operational conditions, allowed for the identification of essential aspects regarding the behavior of these protective layers.

Before exposure to operating cycles, notable differences were observed in the electrochemical behaviors of the various coatings:▪Samples coated with MgZrO_3_-35NiCr (P2) exhibited the most electropositive values for open-circuit potential (Eoc) and corrosion potential (Ecorr), indicating superior electrochemical stability and reduced susceptibility to corrosion.▪Samples coated with Cr_3_C_2_-25(Ni20Cr) (P1) recorded the lowest corrosion current density (icorr) and the highest polarization resistance (Rp), thus confirming their ability to effectively protect the metallic substrate.

After exposure to operational conditions, the results showed that all TBCs contributed to improving the corrosion resistance compared to the base material of the valves. Among them, the ZrO_2_-5CaO (P3) coatings stood out due to the following:▪The most noble electrochemical behavior post-use;▪The most electropositive corrosion potential, indicating superior oxidation resistance;▪The lowest corrosion current density, demonstrating minimal material degradation;▪The highest polarization resistance, confirming the durability of the protective layer under intense thermal and chemical stress conditions.

This behavior can be explained by the structural stability under high-temperature exposure of coating 3, which is completely ceramic, compared to the other two types of coatings, which also contain, in the top coat, certain percentages of metallic components (P1 is completely metallic, while P2 is cermet type). The comparative XRD analysis of the samples showed the formation of oxides of the metals contained in the top coat in P1u and P2u, which confirms the above-mentioned hypothesis.

Morphological analyses performed using scanning electron microscopy (SEM) and energy-dispersive spectroscopy (EDS) indicated the presence of some films formed by the accumulation of combustion byproducts. The role of those residue films regarding the additional protection of TBCs against corrosion is not completely established and needs more investigation.

Based on these results, we can conclude that coating the intake and exhaust valve plates of internal combustion engines with thermal barrier coatings (TBCs) via APS can be beneficial for improving their performance, as it also enhances their resistance to corrosion caused by combustion gases, in addition to the benefits related to reducing heat losses and lowering harmful emissions (as presented in the introductory chapter).

## Figures and Tables

**Figure 1 materials-18-01227-f001:**
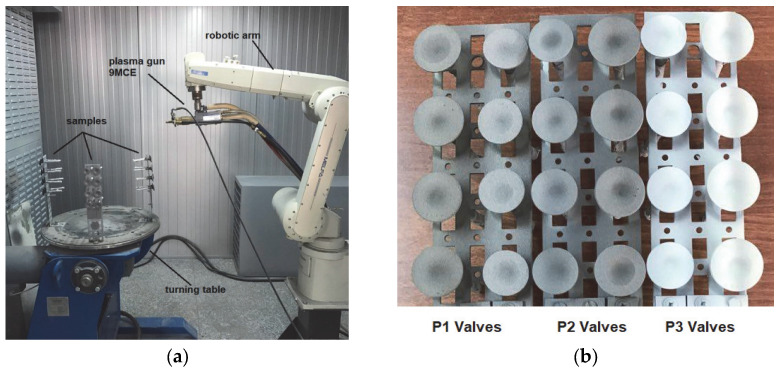
Aspects of the thermal deposition process: (**a**) samples mounted on the deposition stand; (**b**) the appearance of the valves after coating.

**Figure 2 materials-18-01227-f002:**
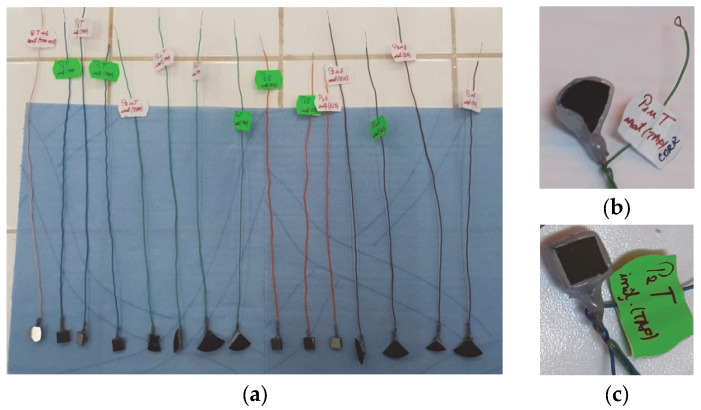
The samples used in the corrosion tests (**a**); mounting details of samples P1u (**b**) and P2 (**c**).

**Figure 3 materials-18-01227-f003:**
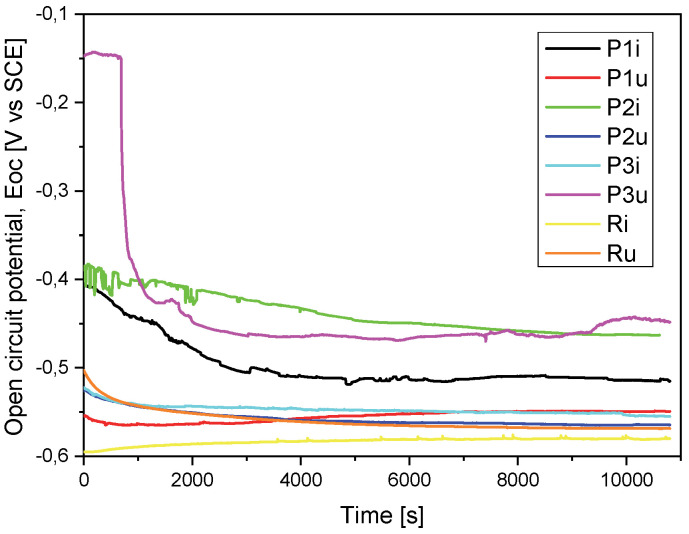
Open-circuit potential evolution (E_oc_) for all the tested samples.

**Figure 4 materials-18-01227-f004:**
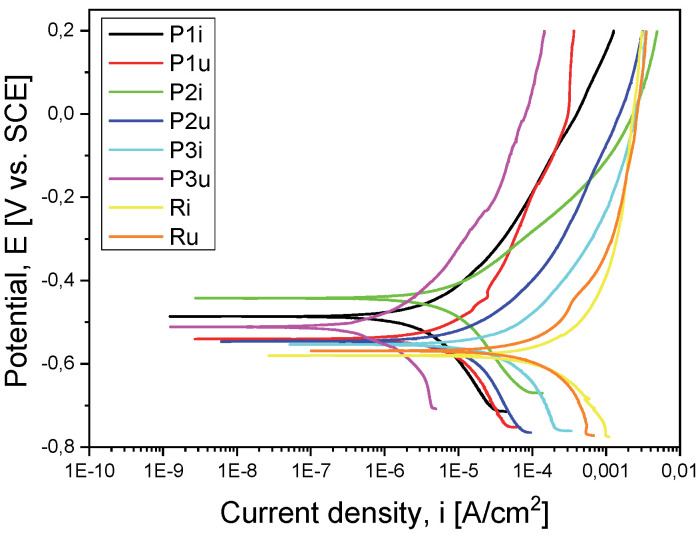
Tafel curves for all the tested samples.

**Figure 5 materials-18-01227-f005:**
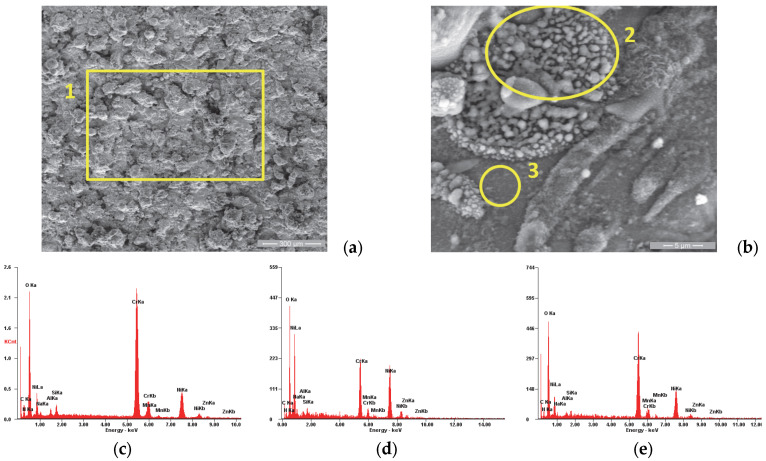
SE images of the P1i sample surface: (**a**) 200×; (**b**) 8000×. EDS spectra for (**c**) area 1, (**d**) area 2 and (**e**) area 3.

**Figure 6 materials-18-01227-f006:**
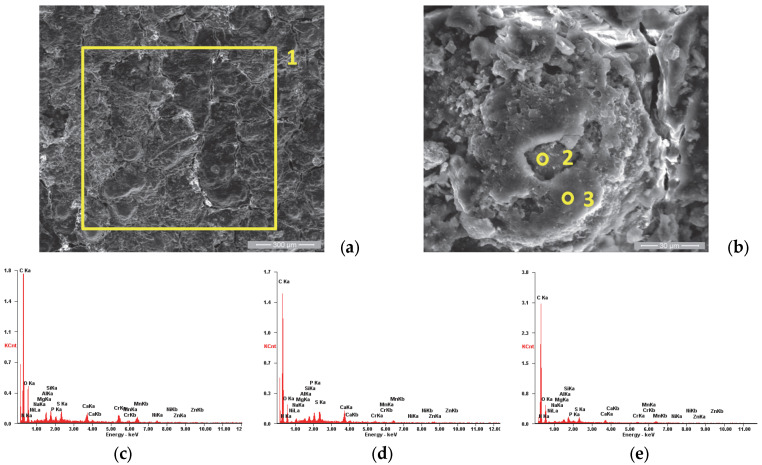
SE images of the P1u sample surface: (**a**) 200×; (**b**) 2000×. EDS spectra for (**c**) area 1, (**d**) point 2 and (**e**) point 3.

**Figure 7 materials-18-01227-f007:**
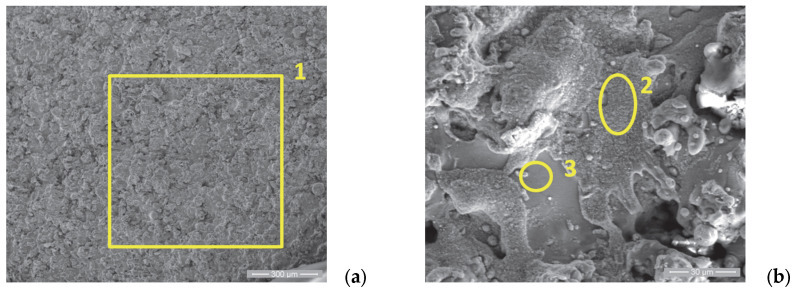
SE images of the P2i sample surface: (**a**) 200×; (**b**) 2000×. EDS spectra for (**c**) area 1, (**d**) area 2 and (**e**) area 3.

**Figure 8 materials-18-01227-f008:**
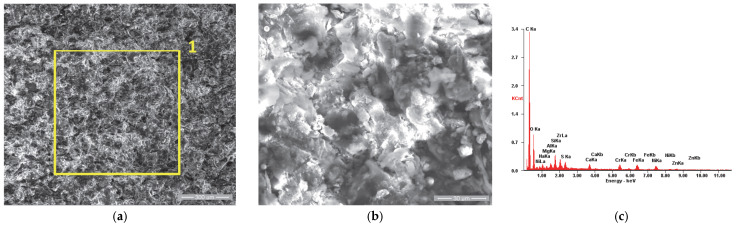
SE images of the P2u sample surface: (**a**) 200×; (**b**) 2000×. EDS spectrum for (**c**) area 1.

**Figure 9 materials-18-01227-f009:**
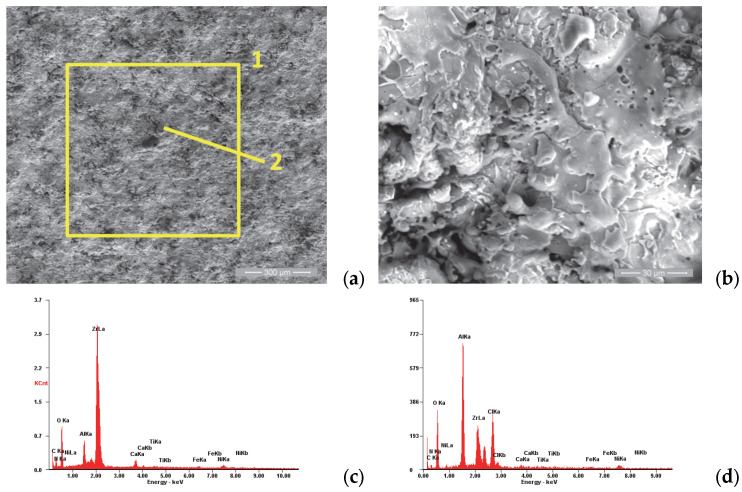
SE images of the P3i sample surface: (**a**) 200×; (**b**) 2000×. EDS spectra for (**c**) area 1 and (**d**) point 2.

**Figure 10 materials-18-01227-f010:**
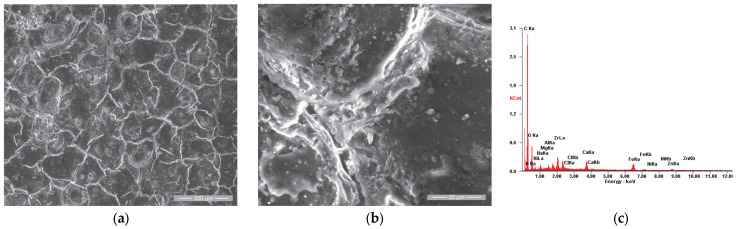
SE images of the P3u sample surface: (**a**) 200×; (**b**) 2000×. (**c**) EDS spectrum for area (**a**).

**Figure 11 materials-18-01227-f011:**
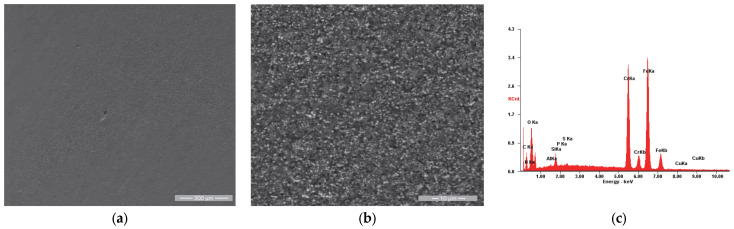
SE images of the Ri sample surface: (**a**) 200×; (**b**) 4000×. (**c**) EDS spectrum for area (**a**).

**Figure 12 materials-18-01227-f012:**
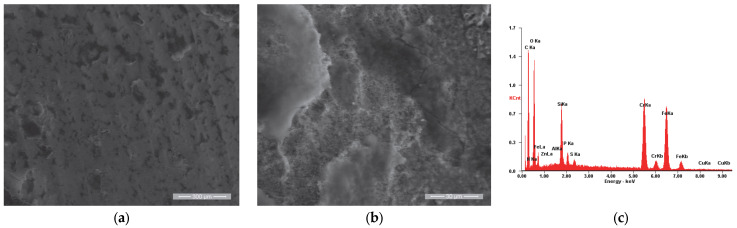
SE images of the Ru sample surface: (**a**) 200×; (**b**) 2000×. (**c**) EDS spectrum for area (**a**).

**Figure 13 materials-18-01227-f013:**
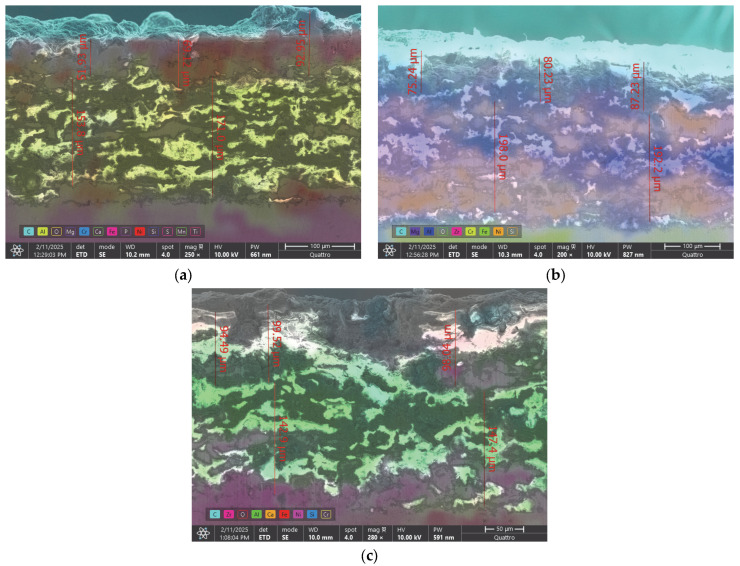
SE images combined with EDS analysis on the cross-sections of (**a**) P1u, (**b**) P2u and (**c**) P3u.

**Figure 14 materials-18-01227-f014:**
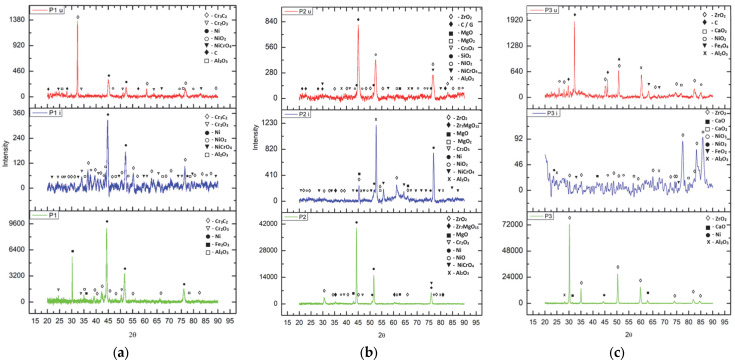
XRD patterns of the three types of TBCs considered: (**a**) Cr_3_C_2_-Ni20Cr; (**b**) MgZrO_3_-35NiCr; (**c**) ZrO_2_-5CaO (with an Al_2_O_3_-30 (Ni20Al) BC in all cases).

**Table 1 materials-18-01227-t001:** Literature survey regarding TBCs used for internal combustion engine coating.

Material (Layer Type)	Spray Method	Component Type/Substrate	Fuel/Testing Procedure	Ref.
NiCoCrAlY (BC) 8YSZ (TC) Ni95Al5 (sealing) Al_2_O_3_ (sealing)	HVAF APS APS SPS	The piston of a single-cylinder light-duty diesel engine/AlSi12	Diesel fuel/CFD simulations	[5]
NiCrAl (BC) 8YSZ (TC)	APS APS	Piston surface from AVL 530 single-cylinder research engine with a custom head designed for high-output diesel engine operation	F-24 (kerosene-based jet fuel)/engine operation	[17]
NiCoCrAIY (BC) 8YSZ (TC)	APS APS	Flame tube liner panel of the V2500 aircraft engine/Hastelloy X	Heating (furnace, 1000, 1100, 1200 and 1300 °C)–rapid cooling (air, 20 °C) cycles	[21]
Metallic powder (BC) YSZ (TC) YSZ+CeO_2_ (TC)	VPS APS APS	Piston top crown/aluminum alloy	D100 + HHO ^1^; Opt.JME20 + HHO/dual-fuel operation	[27]
NiCrAlY (BC) 7YSZ (TC)	HVOF APS	Floating tile surface of the combustion chamber flame tube of aero-engines/nickel-based alloy K417	Heating (furnace, 1150 °C)–cooling (water, 20 °C) cycles	[32]
YSZ (TC) MgZrO_3_ (TC)	- Spray atomization model	In-cylinder combustion simulation/steel	Gasoline; diesel/in-cylinder flow process simulation using Kiva3v software	[34]
NiCoCrAlY (BC) GCSZ ^2^ (TC)	HVOF SPS	Sample plates/ductile iron QT450–10	Heating (furnace, 1000 °C)–cooling (water, 20 °C) cycles	[35]
Ni-Fe-Al (BC1) Al_2_O_3_-Ni-Al (BC2) 8YSZ (50%wt) + Al_2_O_3_ (50%wt) (TC)	APS APS APS	Sample plates/ductile iron QT450–10	Heating (furnace, 1000 °C)–cooling (air, 20 °C) cycles	[36]
La_2_Zr_2_O_7_ (TC)	-	Different coating areas of the combustion chamber of diesel engines	Diesel fuel/KIVA-3V CFD software package default model simulations	[37,38]
MgZrO_3_ (TC)	APS	Top surface and combustion chamber of the piston/aluminum alloy	Gasoline fuel/single-cylinder CI engine operation	[39]
NiCrAl (BC) YBZ ^3^ (TC) YPSZ ^3^ (TC) YSZ ^3^ (TC) MSZ ^3^ (TC) LC ^3^ (TC)	-	Piston surface/Al-Si alloy	Numerical analysis of the steady-state thermo-mechanical behavior of a diesel engine piston	[40]
NiCrAlY (BC) 8YSZ (TC2) YPSZ (TC1) Mullite (TC1) MgZrO_3_ (TC1) La_2_Ce_2_O_7_ (TC1) La_2_Zr_2_O_7_ (TC1)	-	Piston surface/Al-Si alloy	Finite element numerical simulation study of thermal insulation capabilities of double-layer ceramic multi-layer thermal barrier	[41]
YSZ (TC) Cordierite–YSZ (TC)		Piston surface/4140-grade steel	Unsteady heat flux model/fracture-based thermo-mechanics model simulations for reciprocating internal combustion engine	[42]
NiCrAlY (BC) Mullite (TC)	HVOF APS	Mild steel plates	Hot corrosion test at 700 °C (40 wt% Na_2_SO_4_ + 60 wt% V_2_O_5_ salt paste) for 300 h	[43]
GZO ^4^ (TC)	APS	Piston surface/aluminum alloy	Gasoline fuel/single-cylinder research engine operation	[44]
NiCr (BC) 40% Al_2_O_3_ + 30% TiO + 30% Mo (TC)	Electroplating APS	The inner surface of the head of the cylinder and the piston crown of a diesel engine/aluminum alloy	Punnai oil–diesel mixtures/diesel engine operation	[45]
MetcoAmdry997 (BC) CYSZ ^5^ (interlayer) La_2_Zr_2_O_7_ (TC) La_1.4_Yb_0.6_Zr_2_O_7_ (TC) La_1.4_Dy_0.6_Zr_2_O_7_ (TC) La_1.4_Nd_0.6_Zr_2_O_7_(TC)	HVOF APS APS APS APS APS	Surfaces of combustion chamber parts (cylinder head, piston and valve)	Single-cylinder, air-cooled, four-stroke and direct-injection diesel engine operation	[46]
Amdry365-4 (BC 1,2) AMDRY386 (BC 3) APSPoly/YSZ ^6^ (TC) APSPoly/GZO ^6^ (TC) SPS GZO ^6^ (TC)	APS HVAF APS APS SPS	Sample plates/low-carbon steel; Hastelloy-X	Thermal Swing Test for thermal properties (short exposure to a flame produced with an HVAF torch) evaluated via Laser Flash Analysis (LFA)	[47]
NiCrAl (BC) MgZrO_3_ (TC)	APS APS	Diesel engine piston crown/aluminum alloy (silica, copper, chromium, magnesium, etc.)	ANSYS thermal stress analyses/MWM TBRHS 518-V16 direct-injection diesel engine simulation	[48]
NiCrAl (BC) Mg-PSZ (TC) YPSZ (TC)	APS APS APS	Piston top surface/AlSi	Temperature gradient simulation on the piston surface near the crevice volume of the spark ignition engine using the finite element method (FEM)	[49]
NiCrAlY (BC) 8YSZ (TC)	APS APS	Sample plates/A356.0-T7 (7.03% Si, 0.35% Mg, 0.26% Fe, 0.17% Cu, 0.01% Mn, 0.01% Ti and bal. Al)	Flame heating (325 °C, 525 °C and 580 °C) + quenching (40 °C, water) and FE (ABAQUS) simulation of stress distribution under various thermo-mechanical loadings	[50]
NiCrAl (BC) La_2_Ce_2_O_7_ (TC)	APS APS	Piston surface/aluminum alloy	Steady-state thermal analysis using ANSYS R15.0/MWM TBRHS 518-V16 direct-injection diesel engine model	[51]

^1^ D100-D2-quality petro-diesel (D100) and a blend comprising 20% Jatropha oil and 80% D100, named JME20, oxy-hydrogen gas (HHO); ^2^ GCSZ (4 mol% Gd_2_O_3_–16 mol% CeO_2_-ZrO_2_); ^3^ Yttria partially stabilized zirconia (YPSZ), yttria-stabilized zirconia (8YSZ), magnesia-stabilized zirconia (MSZ), lanthanum cerate (LC) and yttrium barium zirconate (YBZ); ^4^ GZO (Gadolinium Zirconate); ^5^ CYSZ (ZrO_2_-24CeO_2_-2,5Y_2_O_3_); ^6^ APSPoly/YSZ (Metco 2460NS), APSPoly/GZO (Metco 6042 + Metco 600 NS-1) and SPS GZO (AE12413).

**Table 2 materials-18-01227-t002:** Encoding of the samples subjected to the electrochemical corrosion tests.

No.	Material Type/State	Encoding
1.	Set 1/initial state	P1i
2.	Set 1/worn	P1u
3.	Set 2/initial state	P2i
4.	Set 2/worn	P2u
5.	Set 3/initial state	P3i
6.	Set 3/worn	P3u
7.	Valve base material/initial state	Ri
8.	Valve base material/worn	Ru

**Table 3 materials-18-01227-t003:** Main electrochemical parameters of the corrosion process.

No.	Sample	E_oc_ (mV)	E_corr_ (mV)	i_corr_ (µA/cm^2^)	|β_c_| (mV/Decade)	β_a_ (mV/Decade)	R_p_ (kΩ × cm^2^)	P (%)	P_e_ (%)
1.	P1i	−515	−487	8.751	485.15	285.36	8.927	1.99	99.20
2.	P1u	−549	−540	21.47	544.76	477.68	5.154	-	-
3.	P2i	−463	−442	13.536	348.25	183.66	3.862	4.41	98.77
4.	P2u	−564	−547	37.36	920.77	295.03	2.6	-	-
5.	P3i	−555	−554	114.582	641.83	337.38	0.839	22.66	89.60
6.	P3u	−448	−511	2.717	668.19	305.58	33.554	-	-
7.	Ri	−580	−580	1102	955.59	1030	0.195	-	-
8.	Ru	−568	−569	481.594	748.31	608.55	0.303	-	-

**Table 4 materials-18-01227-t004:** The chemical composition of P1i, respectively, and P1u sample surfaces after corrosion tests.

Samples	Chemical Elements (wt%)												
	C	O	Na	Mg	Zn	Al	Si	P	S	Ca	Cr	Mn	Ni
P1i-1		19.07	1.44		1.87	1.35	1.83				50.86		23.58
P1i-2		19.37	2.92		5.24	0.79	1.52				25.64		44.53
P1i-3		21.11	0.95		1.43	1.14	1.32				41.9		32.16
P1u-1	56.83	21.97	0.78	0.19	1.88	1.75	2	0.97	2.15	3.14	5.76	0.28	2.31
P1u-2	61.32	16.43	1.2	0.35	4.01	0.89	1.37	2.49	3.19	5.34	1.52	0.55	1.35
P1u-3	63.2	25.86	0.48	0.08	0.85	0.86	2.01	0.59	1.89	1.91	1.23	0.31	0.74

**Table 5 materials-18-01227-t005:** The chemical composition of the P2i, respectively, and P2u sample surfaces after corrosion tests.

Samples	Chemical Elements (wt%)												
	C	O	Na	Mg	Zn	Al	Si	Zr	S	Ca	Cr	Ni	Fe
P2i-1		24.62	0.34	1.42	0.48	0.48	1.8	4.28	0.36	0.33	32.95	33.43	
P2i-2		23.42	0.42	0.42	0.33	0.33	0.4	1.25	0.32	0.35	44.45	28.63	
P2i-3		24.36	0.17	0.38	0.3	0.3	0.74	1.34	0.26	0.46	32.21	39.78	
P2u-1	52.39	19.65	1.48	0.46	1.89	1.09	2.76	4.35	1.62	1.61	3.26	5.00	4.42

**Table 6 materials-18-01227-t006:** The chemical composition of the P3i, respectively, and P3u sample surfaces after corrosion tests.

Samples	Chemical Elements (wt%)											
	C	O	Na	Mg	Zn	Al	Zr	Cl	Ca	Ti	Ni	Fe
P3i-1		27.71				4.68	59.33		2.66	0.68	3.60	1.34
P3i-2		31.78				24.21	21.40	16.13	0.95	0.47	3.77	1.29
P3u-1	51.95	21.72	1.75	0.25	3.85	0.91	7.45	0.11	4.17		0.58	7.26

**Table 7 materials-18-01227-t007:** The chemical composition of the Ri, respectively, and Ru sample surfaces after corrosion tests.

Samples	Chemical Elements (wt%)									
	C	O	Zn	Al	Si	P	S	Cr	Fe	Cu
Ri		5.85		0.47		0.34	0.38	33.2	56.99	0.65
Ru	23.22	20.39	0.21	0.22	6.79	1.38	0.59	19.31	27.24	0.64

## Data Availability

The original contributions presented in this study are included in the article. Further inquiries can be directed to the corresponding author(s).

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
