# Peer review of "Assessment of the Corrosion Resistance of Thermal Barrier Coatings on Internal Combustion Engine Components"

_materials, 2025, doi:10.3390/ma18061227_

Round 1
Reviewer 1 Report
Comments and Suggestions for Authors
The authors present the study of the various types of thermal barrier coatings in real samples of the valves and in simulated combustion environmental conditions. The article is well organized and the methods are appropriate for this kind of study. Besides electrochemical techniques, the authors used various methods in order to obtain data related to structure and morphology. The methodology is well chosen. The reported literature reflects the scope of the subject.
In order to improve the manuscript some changes should be made.
The authors presented in Figure 2 the construction of the samples. However, it is not clear how they performed contact between samples and wire and how they insulated that contact, or the whole electrode body.
Also, as the pH of the electrolyte is 3, it should be emphasis real form of species in electrolyte.
Statement in lines 295-296: "….a more electropositive potential (-463 mV) and consequently a more electrochemical character and better corrosion behavior." is not clear.
What is means by "better corrosion behavior".
Also, the authors should comment on the parameters obtained by corrosion measurements related to the initial and worn state, as in some samples, drastic improvement is observed in the worn samples. For example, P3i vs P3u, despite some anodic shift of the corrosion potential. On the contrary, the cathodic shift of the potential for worn samples is observed in the ample P2u but with an increase in the corrosion current.
Author Response
- Summary
Estimeed Reviewer, thank you very much for taking the time to review this manuscript. Please find the detailed responses below and the corresponding revisions/corrections highlighted with yellow in the re-submitted file.
- Point-by-point response to Comments and Suggestions for Authors
The authors present the study of the various types of thermal barrier coatings in real samples of the valves and in simulated combustion environmental conditions. The article is well organized and the methods are appropriate for this kind of study. Besides electrochemical techniques, the authors used various methods in order to obtain data related to structure and morphology. The methodology is well chosen. The reported literature reflects the scope of the subject. In order to improve the manuscript some changes should be made.
Comments 1: The authors presented in Figure 2 the construction of the samples. However, it is not clear how they performed contact between samples and wire and how they insulated that contact, or the whole electrode body.
Response 1: Thank you for the valuable comments and questions. They were very helpful in improving the quality of the paper and we hope that the answer is according to your expectations. We have revised the paper with the following statement to emphasize this point: `` Prior to the corrosion tests the samples were prepared in order to ensure electrical contact followed by an isolation of the electrical contact and of the entire sample, leaving only the surface of interest to be exposed to the electrolyte. Thus, to ensure electrical contact, an insulated Cu wire was micro-welded on the surface opposite the thermal sprayed surfaces. The electrical contact and the entire sample (excepting the surface of interest) were then covered with silicone (Figure 2). Subsequently, using macroscopic images of each embedded samples and ImageJ software, the area of the non-silicone coated surfaces was measured and then used for the calculation of the electrochemical parameters.” (lines 232 – 239)
Comments 2: Also, as the pH of the electrolyte is 3, it should be emphasis real form of species in electrolyte.
Response 2: Thank you for the valuable comments and questions. They were very helpful in improving the quality of the paper and we hope that the answer is according to his expectations. The ionic and reactive species that form the electrolyte are specified on line 255.
Comments 3: Statement in lines 295-296: "….a more electropositive potential (-463 mV) and consequently a more electrochemical character and better corrosion behavior." is not clear.
Response 3: We are grateful for this observation. It was very helpful in improving the quality of the paper and we hope that the answer is according to your expectations. In order to draw comparisons between the potential values, it is necessary to utilize the electrochemical series, in which the values in the upper part (electropositive) possess a more noble character from the electrochemical point of view. The same principle can be employed when comparing values resulting from experiments. As a result of this observation, the following modifications have been made to the reported sentence: “….a more electropositive potential (-463 mV) and consequently a more noble electrochemical character and possibly a better corrosion behavior.” (lines 309 – 310)
Comments 4: What is means by "better corrosion behavior".
Response 4: Thank you for the valuable comments and questions. They were very helpful in improving the quality of the paper and we hope that the answer is according to your expectations. As presented, the aim of this work is to evaluate how coatings influence the degree of valve corrosion in the simulated gas environment produced by fuel oil combustion. In order to distinguish between the effect of the three coatings, we have used the term ‘corrosion behaviour’ and have appealed to the comparative and superlative forms of the adjective ‘good’”. In the context of corrosion test results, the term 'better corrosion behaviour' is employed exclusively in the context of comparisons between experimental samples, with the objective of highlighting the performance of a particular material in relation to the electrochemical parameters under consideration. Consequently, the term 'better corrosion behaviour' may, depending on the specific parameters under consideration, be regarded as analogous to the investigated materials.
Comments 5: Also, the authors should comment on the parameters obtained by corrosion measurements related to the initial and worn state, as in some samples, drastic improvement is observed in the worn samples. For example, P3i vs P3u, despite some anodic shift of the corrosion potential. On the contrary, the cathodic shift of the potential for worn samples is observed in the ample P2u but with an increase in the corrosion current.
Response 5: Thank you for the valuable comments and questions. They were very helpful in improving the quality of the paper and we hope that the answer is according to your expectations. The comments regarding the improvement of the behavior of the worn samples were included in the Conclusions chapter, which was completely redone.
- Response to Comments on the Quality of English Language
Point 1: The English is fine and does not require any improvement.
Response 1: We appreciate your valuable feedback, and we have meticulously reviewed the entire manuscript to guarantee both grammatical precision and academic rigor. Your comments have been instrumental in enhancing the quality of the paper, and we are confident that the revised version now meets the grammar and academic requirements.

Reviewer 2 Report
Comments and Suggestions for Authors
This manuscript discusses the corrosion resistance of three different TBCs on the internal combustion engine valve plates. The authors performed XRD, SEM and PDP measurements to evaluate the materials properties. The research design is inadequate and the contents in the introduction part should be focused on the research topic rather than being more general. There are several questions remain and couldn’t conclude the outcome. The journal in its present form is not suitable for publication and the following comments must be addressed before the author comments on publication in Materials Journal.
1. Introduction part – not focused and should reduce the content – the authors discusses GHC emissions and the way to mitigate them, basic structure of TBCs, literature survey on TBC coated combustion engine coatings, relation between TBC coating and CO emissions, the need for TBCs on combustion chambers, NOx emissions, compatibility with biofuels (don’t know why the authors discussed here), and finally the authors approach on coatings in intake and exhaust valve plates. Some of the questions related to this manuscript remain. What are the materials proposed for these TBC applications and what research has undergone so far? What is the research gap on these materials, how process methods can support authors approach, and finally authors approach.
2. Why Al2O3 - 30 (Ni20Al) powder was used for preparing bond coat than conventional MCrAlY.
3. On what basis does the author choose P1, P2 and P3 composition? Have the authors optimized this composition before?
4. What is the thickness of the bond coat and top coat and what are the APS process parameters used for depositing this type of TBC materials?
5. Corrosion measurement – how these samples were masked and immersed in the electrolyte. What is the exposure area?
6. Figure 12- the authors marked noise level as a peak and this is not correct, especially 12C and 12b and 12a coated components. What is the reason for orientation changes. I doubt the thickness of layers and their adhesion. Provide cross-sectional SEM image of all three coated components or stylus profilometer data.
7. Conclusion – look like a summary and extended version of discussion- should be revised
8. How did the author confirm the tribofilm formation influence the corrosion properties. I couldn’t see any differences from PDP graphs.
9. Provide optical images of corroded samples (after corrosion) to see the type of corrosion that occurred.
10. Table 7 – provide chemical composition of the coating and the materials used for coating also. I doubt on the decomposition of the layers.
Author Response
Response to Reviewer 2 Comments
- Summary
Estimeed Reviewer, we would like to express our heartfelt gratitude for the time and effort you have dedicated to reviewing our paper and providing us with your valuable comments and suggestions. We are sincerely grateful for your thoughtful contribution and support throughout the review process.
- Point-by-point response to Comments and Suggestions for Authors
This manuscript discusses the corrosion resistance of three different TBCs on the internal combustion engine valve plates. The authors performed XRD, SEM and PDP measurements to evaluate the materials properties. The research design is inadequate and the contents in the introduction part should be focused on the research topic rather than being more general. There are several questions remain and couldn’t conclude the outcome. The journal in its present form is not suitable for publication and the following comments must be addressed before the author comments on publication in Materials Journal.
Comments 1: Introduction part – not focused and should reduce the content – the authors discusses GHC emissions and the way to mitigate them, basic structure of TBCs, literature survey on TBC coated combustion engine coatings, relation between TBC coating and CO emissions, the need for TBCs on combustion chambers, NOx emissions, compatibility with biofuels (don’t know why the authors discussed here), and finally the authors approach on coatings in intake and exhaust valve plates. Some of the questions related to this manuscript remain. What are the materials proposed for these TBC applications and what research has undergone so far? What is the research gap on these materials, how process methods can support authors approach, and finally authors approach.
Response 1: Thank you for the valuable comments and questions. They were very helpful in improving the quality of the paper and we hope that the answer is according to your expectations. We have revised the content of the introductory chapter and removed some of the explanations of the analyzed data from the specialized literature, as they made it more difficult to understand the purpose of this study (lines 44-45, 102-103, 121-127 and 414-147 in the original manuscript). Regarding the questions related to the selection of materials for this study, I have added clarifications on this topic in Chapter 2.1 (lines 171-184 in the revised version). The research gap is related, firstly, to the fact that no metallic or cermet powders were studied for those applications, secondly to the fact that most of the TBCs coated surfaces were the ones from the cylinder head and piston (not from the exhaust / inlet valves as in this study) and thirdly to the introduction of the corrosion environment that simulated the combustion gases.
Comments 2: Why Al2O3 - 30 (Ni20Al) powder was used for preparing bond coat than conventional MCrAlY.
Response 2: Thank you for your valuable advice, which guided us in adding explanations where our previous descriptions were lacking. In accordance with your suggestion, we have revised the manuscript to include the following statement: "As presented in the specialized literature [55], the presence of a thermally grown oxide (TGO) layer with a thickness of 2-3 microns is sufficient to protect the substrate from the effects of thermal shock. Furthermore, it has been demonstrated that uncontrolled oxidation of the bond coat (BC), along with TGO growth, can generate excessive internal stresses, leading to a reduction in interlayer adhesion, which may ultimately result in delamination or spallation [56]. Based on these observations, we have chosen a BC formulation that already contains alumina (Alâ‚‚O₃) as a thermal barrier component, while being coupled with a metallic matrix to ensure substrate adhesion. The selection of this powder was in accordance with the manufacturer’s recommendations [57].`` (lines 169-178)
Comments 3: On what basis does the author choose P1, P2 and P3 composition? Have the authors optimized this composition before?
Response 3: Thank you for your valuable advice, which guided us in adding explanations where our previous descriptions were lacking. In accordance with your suggestion, we have revised the manuscript to include the following statement: ``For the top coat, a representative powder was selected from each of the three major categories of powders used in thermal spray processes: metallic powders (P1), cermet powders (P2), and ceramic powders (P3). The selection was based on references from the specialized literature (see Table 1) and the manufacturer’s specifications [58-60].`` (lines 179-182)
Comments 4: What is the thickness of the bond coat and top coat and what are the APS process parameters used for depositing this type of TBC materials?
Response 4: Thank you for your valuable observation, which guided us in adding explanations where our previous descriptions were lacking. In accordance with your suggestion, we have revised the manuscript to include the following statement: ``The spraying parameters used to obtain the coatings were as follows: N2 and H2 flow was 3,4 Barr (same for all the powders); the Ar (carrier gas) flow was 5,66 NLPM for BC and P1, respectively 5,5 NLPM for P2 and P3; the voltage was 70÷80 V for BC, respectively 70÷80 V for P1, P2 and P3; the intensity was 500 A for all the powders; the spraying distance was 153 mm for BC, 90 mm for P1, 120 mm for P2 and 110 mm for P3. The feed rate was also different for each powder: 68 g/min for BC, 91 g/min for P1, 65 g/min for P2 and 87 g/min for P3. After the deposition was finished, one sample from each batch was metallographically prepared for cross-section analysis, and was concluded that the thickness of the bond coat had a medium value of 170 µm and the one of the top coats was 75 µm..`` (lines 203-212)
Comments 5: Corrosion measurement – how these samples were masked and immersed in the electrolyte. What is the exposure area?
Response 5: Thank you for the valuable comments and questions. They were very helpful in improving the quality of the paper and we hope that the answer is according to your expectations. In accordance with your suggestion, we have revised the manuscript and included the following statement: ``Prior to the corrosion tests the samples were prepared in order to ensure electrical contact followed by an isolation of the electrical contact and of the entire sample, leaving only the surface of interest to be exposed to the electrolyte. Thus, to ensure electrical contact, an insulated Cu wire was micro-welded on the surface opposite the thermal sprayed surfaces. The electrical contact and the entire sample (excepting the surface of interest) were then covered with silicone (Figure 2). Subsequently, using macroscopic images of each embedded samples and ImageJ software, the area of the non-silicone coated surfaces was measured and then used for the calculation of the electrochemical parameters.`` (lines 232-240)
Comments 6a: Figure 12- the authors marked noise level as a peak and this is not correct, especially 12C and 12b and 12a coated components. What is the reason for orientation changes.
Response 6a: Thank you for your valuable observation, which guided us in adding explanations where our previous descriptions were lacking. In accordance with your suggestion, we have revised the manuscript and included the COD number for all the identified peaks along with the discussions regarding their presence.
Comments 6b: I doubt the thickness of layers and their adhesion. Provide cross-sectional SEM image of all three coated components or stylus profilometer data.
Response 6b: Thank you for your valuable observation, which guided us in adding explanations where our previous descriptions were lacking. In accordance with your suggestion, we have revised the manuscript to include Figure 13. SE images combined with EDS analysis on the cross-sections of: a) P1u; b) P2u, c) P3u and the following statement: ``The cross-sections of samples extracted from the worn valve plates were metallo-graphically prepared and analyzed by SEM method, on a FEG electron microscope (Quattro C, Thermo Fisher, Holland, 2024) using the High Vacuum mode, ETD detector, 10kV electron beam acceleration and a 10 mm working distance, as presented in Figure 12a,b and c. With the help of the EDS incorporated system, the main chemical elements from the cross-sections were emphasized on the chemical distribution maps, that were superimposed on the secondary electrons images. As a result, in Figure 12 a is visible the distribution of the main chemical elements that form the substrate (Fe - pink), the bond coat (Al – yellow, Ni – red) and the top coat (Cr – blue, Ni – red) in the case of P1u sample. The superior surface is colored in light blue which represents the carbon, this being also a proof of the residue film presence. In the same manner, we have analyzed the cross-section of sample P2u and observed the presence of Fe (green) in the substrate, of Al (blue) and Ni (orange) in the bond coat, and the presence of Mg (purple), Zr (pink), Cr (yellow) and Ni (orange) in the top coat. The superior side of the sample was also colored light blue, which represents the presence of carbon. Figure 12 c represents the cross-section of the sample P3u, with Fe (red) in substrate, with Al (green) and Ni (pink) in the bond coat, with Zr (pink), Ca (orange) and Si (blue) in the top coat and with carbon on the surface of the coating (light-blue). The oxygen was not marked on the distribution map because its presence could introduce interpretation errors. In each of the three cases it can be observed that the TBC layers maintained their integrity throughout the entire cross-section, and the adhesion to the substrate was not negatively influenced by the corrosive environment or the high temperatures to which it was exposed during the 36 hours of continuous engine operation.`` (lines 455-480)
Comments 7: Conclusion – look like a summary and extended version of discussion- should be revised
Response 5: Thank you for your valuable observation, which guided us in adding explanations where our previous descriptions were lacking. In accordance with your suggestion, we have completely revised the Conclusion chapter.
Comments 8: How did the author confirm the tribofilm formation influence the corrosion properties. I couldn’t see any differences from PDP graphs.
Response 5: Thank you for the valuable comments and questions. They were very helpful in improving the quality of the paper and we hope that the answer is according to your expectations. In accordance with your observation, we considered that the data currently available do not allow us to conduct extensive discussions regarding the extent to which this film, formed due to combustion residues, can decisively influence the overall corrosion behavior of the TBC layers after operation. For this reason, we have removed these observations from the initial manuscript.
Comments 9: Provide optical images of corroded samples (after corrosion) to see the type of corrosion that occurred.
Response 9: Thank you for the valuable comments and questions. They were very helpful in improving the quality of the paper and we hope that the answer is according to your expectations. Due to the complexity of the materials under investigation, detailed investigations are necessary to accurately highlight the types of corrosion that have occurred as a result of the tests. Moreover, highlighting the types of corrosion that may occur after testing was not one of the objectives of this study. Consequently, the test parameters were selected (linear polarization curves from -200 mV (vs. OCP) to +200 mV (vs. OCP) - Tafel curves) and due to the low applied potential, no visible corrosion phenomena were observed. For the induced corrosion process to be stronger and thus for traces of the corrosion process to appear on the surface of the samples, the applied potential had to be at least ± 1000 mV.
We realized the optical images of the corroded samples (40x, NexiusZoom EVO 0.65x to 5.5x Stereo Microscope, Euromex, 2022) as presented below, but we didn’t insert them into the revised version of the paper because they are not representative for this study, as explained before.
(a) |
(b) |
(c) |
(d) |
(e) |
(f) |
(g) |
(h) |
The optical aspect of corroded sample: a) P1i; b) P2i; c) P3i; d) Ri; a) P1u; b) P2u; c) P3u; d) Ru.
Comments 10: Table 7 – provide chemical composition of the coating and the materials used for coating also. I doubt on the decomposition of the layers.
Response 10: Thank you for your observations, which enhance the reliability of this paper. The chemical composition of the coatings is presented already in the chapter 2.1.
- Response to Comments on the Quality of English Language
Point 1: The English is fine and does not require any improvement.
Response 1: We appreciate your valuable feedback, and we have meticulously reviewed the entire manuscript to guarantee both grammatical precision and academic rigor. Your comments have been instrumental in enhancing the quality of the paper, and we are confident that the revised version now meets the grammar and academic requirements.

Reviewer 3 Report
Comments and Suggestions for Authors
The authors have evaluated the corrosion resistance in an environment equivalent to that generated by the combustion gases for three types of thermal barrier coatings (TBC) deposited by atmospheric plasma spray on the intake/exhaust valves of a gasoline internal combustion engine, both before and after their use in operation. Please address the following comments:
- Please mention all the spray coating process parameters for the BCs deposited on all the valve plates.
- Please mention all the spray coating process parameters for the top coats deposited on all the valve plates.
- Please correct ``In-situ``, in line 207.
- Please correct „in operation”, in line 208.
- Please mention the number of measurements taken for the chemical composition summarized in table 4 and confirm it was area measurements and not the point measurements.
- Please modify table 4 by replacing all values containing ‘,’ with ‘.’.
- Authors mentions ‘Figure 6a shows the surface of the P1 sample after operation tests and the presence of the tribofilm’. Please provide the XPS analysis for this confirming the formation of the relevant tribofilms, and what type of tribofilms were formed? Explain in detail how it was formed with relevant scientific explanation and references as EDS is not right technique to confirm tribofilm formation, but XPS is the right one.
- Please include the elemental mapping images for figure 5 and 6, showing the distribution of various elements.
- Please mention the number of measurements taken for the chemical composition summarized in table 5 and confirm it was area measurements and not the point measurements.
- Please modify table 5 by replacing all values containing ‘,’ with ‘.’.
- Authors mentions ‘In Figure 8 a, b is presented the morphology of the P2u sample surface, where a thick of tribofilm with a frail appearance is visible’. Please provide the XPS analysis for this confirming the formation of the relevant tribofilms, and what type of tribofilms were formed? Explain in detail how it was formed with relevant scientific explanation and references as composition measurements with EDS is not right technique to confirm tribofilm formation.
- Please include the elemental mapping images for figure 8 and 10, showing the distribution of various elements.
- Authors mentions ‘Figure 10 shows the presence of the tribofilm produced after the accumulation of combustion residues, deposited on the surface of the sample and cracked as a result of the oxidation processes at high temperature, as shown in the detail of Figure 10b’. Please provide the XPS analysis for this confirming the formation of the relevant tribofilms, and what type of tribofilms were formed? Explain in detail how it was formed with relevant scientific explanation and references as composition measurements with EDS is not right technique to confirm tribofilm formation.
- Please modify table 6 by replacing all values containing ‘,’ with ‘.’.
- Authors mentions ‘Similar to the other three samples, the appearance after use in the engine operation, shown in Figure 12a, was analyzed. However, it is observed that, in the case of this sample, the deposited tribofilm has the smallest thickness, being in fact in the form of flakes arranged on the entire surface of the valve plate. This aspect is highlighted in Figure 12 b, where the distribution limit of some flakes in the produced tribofilm is captured’. Please provide the XPS analysis for this confirming the formation of the relevant tribofilms, and what type of tribofilms were formed? Explain in detail how it was formed with relevant scientific explanation and references as composition measurements with EDS is not right technique to confirm tribofilm formation.
- Authors mentions that ‘However, it is observed that, in the case of this sample, the deposited tribofilm has the smallest thickness, being in fact in the form of flakes arranged on the entire surface of the valve plate.’ Please measure the thickness of tribofilm to validate this claim, as generally the tribfilms are hardly few nm thick.
- Please modify table 7 by replacing all values containing ‘,’ with ‘.’.
- Please provide the JCPDS reference number for ‘Figure 12. XRD patterns of the three types of TBC considered: a) Cr3C2 - Ni20Cr; b) MgZrO3 - 35NiCr; c) ZrO2 - 5CaO (with an Al2O3- 30 (Ni20Al) BC in all the cases)’ for all compounds.
- Authors mentions that ‘In contrast to the three samples previously analyzed, it should be emphasized that in the case of the control sample, the presence of the carbon element is observed in a much smaller percentage, which confirms the appearance of the thin, uneven tribofilm distributed on the surface of the valve’ I again request to confirm the presence of tribfilms and their thickness with XPS as it is the right technique to identify the formation of tribofilms even when the thickness of such films is less than 10nm and not XRD, as XRD is bulk characterization technique used for phase identification.
- Authors mentions ‘the diffractograms obtained being comparatively presented in Figure 13.’, please correct it, as it should be figure 12.
- Please modify the conclusion with the actual experimental values, the coatings composition/architecture, why they are better compared to other materials used in this experiment rather than providing the generalized statements of the observed trend.
Author Response
Response to Reviewer 3 Comments
- Summary
Thank you very much for taking the time to review this manuscript. Please find the detailed responses below and the corresponding revisions/corrections highlighted with yellow color in the re-submitted files
- Point-by-point response to Comments and Suggestions for Authors
Comments 1: Please mention all the spray coating process parameters for the BCs deposited on all the valve plates.
Response 1: Thank you for your valuable advice, which guided us in adding explanations where our previous descriptions were lacking. In accordance with your suggestion, we have revised the manuscript to include the following statement: "The spraying parameters used to obtain the coatings were as follows: N2 and H2 flow was 3,4 Barr (same for all the powders); the Ar (carrier gas) flow was 5,66 NLPM for BC and P1, respectively 5,5 NLPM for P2 and P3; the voltage was 70÷80 V for BC, respectively 70÷80 V for P1, P2 and P3; the intensity was 500 A for all the powders; the spraying distance was 153 mm for BC, 90 mm for P1, 120 mm for P2 and 110 mm for P3. The feed rate was also different for each powder: 68 g/min for BC, 91 g/min for P1, 65 g/min for P2 and 87 g/min for P3.” (lines 203-209).
Comments 2: Please mention all the spray coating process parameters for the top coats deposited on all the valve plates.
Response 2: Thank you for your valuable advice, which guided us in adding explanations where our previous descriptions were lacking. In accordance with your suggestion, we have revised the manuscript to include the spray parameters for the three types of top coats (referred as P1, P2 and P3) in the paragraph mentioned above as response to comment 1.
Comments 3: Please correct ``In-situ``, in line 207.
Response 3: Thank you for your advice on improving the wording of the paper. Following your suggestion, we have revised the expression „In-situ”.
Comments 4: Please correct „in operation”, in line 208.
Response 4: Thank you for your advice on improving the wording of the paper. Following your suggestion, we have revised the expression „in operation”.
Comments 5: Please mention the number of measurements taken for the chemical composition summarized in table 4 and confirm it was area measurements and not the point measurements.
Response 5: Thank you for the valuable comments and questions. They were very helpful in improving the quality of the paper and we hope that the answer is according to the expectations. For the chemical analysis presented in Table 4, the scan was acquired three times. Whether the area analyzed in terms of chemical composition is a surface or a point it is indicated in Figures 6 (sample P1i) and 7 (sample P1u) completed with the spectra emitted from the EDS analysis.
Comments 6: Please modify table 4 by replacing all values containing ‘,’ with ‘.’.
Response 6: Thank you for your advice on improving the wording of the paper. Following your suggestion, I have revised the table 4 by replacing all values containing ‘,’ with ‘.’.
Comments 7: Authors mentions ‘Figure 6a shows the surface of the P1 sample after operation tests and the presence of the tribofilm’. Please provide the XPS analysis for this confirming the formation of the relevant tribofilms, and what type of tribofilms were formed? Explain in detail how it was formed with relevant scientific explanation and references as EDS is not right technique to confirm tribofilm formation, but XPS is the right one.
Response 7: Thank you for the valuable comments and questions. They were very helpful in improving the quality of the paper and we hope that the answer is according to the expectations. For the beginning, I would like to emphasize that the aim of this study was not to highlight the influence of this film that forms on the surface of the valves in case of use in real engine operation but to evaluate how TBC coatings influence the corrosion resistance in a simulated corrosive environment, similar to the situation of gases emitted during fuel combustion. For these reasons, the discussion of this phenomenon is intended to be more analytical - comparative rather than exhaustive.
In order to identify this film, we turned to the literature (in particular the studies of R. Flo et al.) in which we found references to this phenomenon, under conditions of experiments similar to those addressed in our study. In these studies, the term used is ``tribofilm`` (which is the reason why we have taken it for this study). In light of your observations, we have reanalyzed the bibliographical sources and considered it appropriate to change the name of tribofilm to ``residual film``, to avoid any misunderstanding generated by the prefix ``tribo-`` (which suggests the study of wear phenomena). Thus, we replaced in the revised version of the article the term tribofilm with ``residue film``.
The evaluation with EDS is validated by the cited specialized studies, in which the evidence of these films has been performed with this method.
Regarding the explanations of its formation, based on the cited references we have introduced the following paragraphs: ``When analyzing the surface of sample P1u it was observed a major change in its appearance with the formation of a film, as shown in Figure 6a. The elemental chemical analysis carried out on area 1 showed that it is rich in carbon, as presented on the EDS spectra emitted (Figure 6c) and in Table 4.`` (lines 361-364)
``Based on several studies in the literature [63 -67] that analyze the formation of residual layers ("tribofilm") after exposure of internal combustion engine components to real operating conditions, we concluded that the same phenomenon occurred in our study. The formation of those films was studied in detail by R. Flo et al.[63], who have shown that they are due to the deposition of combustion residues on the surfaces with which the combustion gases come into contact.`` (lines 368-373)
Comments 8: Please include the elemental mapping images for figure 5 and 6, showing the distribution of various elements.
Response 8: Thank you for your advice, which has been instrumental in strengthening the scientific quality of the manuscript. Following your guidance, we have added for each of the analysed areas/points the spectra emitted from the EDS analysis (the distribution map of the detected elements was not generated at that moment), which is why the figures have been completed with points c, d and e. After all the revisions, the figures are: Figure 6 a,b,c,d,e for sample P1i and Figure 7 a,b,c,d,e for sample P1u.
Comments 9: Please mention the number of measurements taken for the chemical composition summarized in table 5 and confirm it was area measurements and not the point measurements.
Response 9: Thank you for the valuable comments and questions. They were very helpful in improving the quality of the paper and we hope that the answer is according to the expectations. For the chemical analysis presented in Table 5, the scan was acquired three times. Whether the area analysed in terms of chemical composition is a surface or a point it is indicated in Figures 8 (sample P2i) and 9 (sample P2u) completed with the spectra emitted from the EDS analysis.
Comments 10: Please modify table 5 by replacing all values containing ‘,’ with ‘.’.
Response 10: Thank you for your advice on improving the wording of the paper. Following your suggestion, we have revised the table 5 by replacing all values containing ‘,’ with ‘.’.
Comments 11: Authors mentions ‘In Figure 8 a, b is presented the morphology of the P2u sample surface, where a thick of tribofilm with a frail appearance is visible’. Please provide the XPS analysis for this confirming the formation of the relevant tribofilms, and what type of tribofilms were formed? Explain in detail how it was formed with relevant scientific explanation and references as composition measurements with EDS is not right technique to confirm tribofilm formation.
Response 11: Thank you for the valuable comments and questions. They were very helpful in improving the quality of the paper. In accordance with your suggestion, we have revised the manuscript to include the following statement: ``Figure 8 a, b presents the morphology of the P2u sample surface, where a residue film with a frail appearance is visible. According to the R. Flo el al. studies regarding the ``tribofilms`` dynamics [64,65], we could consider that the residue film is in its equilibri-um phase, almost fully covering the surface.`` (lines 394-397)
Comments 12: Please include the elemental mapping images for figure 8 and 10, showing the distribution of various elements.
Response 12: Thank you for your advice, which has been instrumental in strengthening the scientific quality of the manuscript. Following your guidance, we have added for each of the analysed areas/points the spectra emitted from the EDS analysis (the distribution map of the detected elements was not generated at that moment), which is why the figures have been completed with more points. After all the revisions, the figures are: Figure 8 a,b,c,d,e for sample P2i, Figure 9 a,b,c for sample P2u, Figure 10 a,b,c,d for sample P3i, Figure 11 a,b,c for sample P3u, Figure 12 a,b,c for sample Ri, Figure 13 a,b,c for sample Ru.
Comments 13: Authors mentions ‘Figure 10 shows the presence of the tribofilm produced after the accumulation of combustion residues, deposited on the surface of the sample and cracked as a result of the oxidation processes at high temperature, as shown in the detail of Figure 10b’. Please provide the XPS analysis for this confirming the formation of the relevant tribofilms, and what type of tribofilms were formed? Explain in detail how it was formed with relevant scientific explanation and references as composition measurements with EDS is not right technique to confirm tribofilm formation.
Response 13: Thank you for the valuable comments and questions. They were very helpful in improving the quality of the paper. In accordance with your suggestion, we have revised the manuscript to include the following statement: ``Figure 10 a,b shows the presence of the residue film produced after the accumulation of combustion residues, deposited on the surface of the valve plate P3u. The film is interrupted by cracks, with an aspect characteristic of the breakdown stage of the ``tribofilms`` dynamics according to R. Flo el al. [64,65].`` (lines 409-412)
Comments 14: Please modify table 6 by replacing all values containing ‘,’ with ‘.’.
Response 14: Thank you for your advice on improving the wording of the paper. Following your suggestion, I have revised table 6 by replacing all values containing ‘,’ with ‘.’.
Comments 15: Authors mentions ‘Similar to the other three samples, the appearance after use in the engine operation, shown in Figure 12a, was analyzed. However, it is observed that, in the case of this sample, the deposited tribofilm has the smallest thickness, being in fact in the form of flakes arranged on the entire surface of the valve plate. This aspect is highlighted in Figure 12 b, where the distribution limit of some flakes in the produced tribofilm is captured’. Please provide the XPS analysis for this confirming the formation of the relevant tribofilms, and what type of tribofilms were formed? Explain in detail how it was formed with relevant scientific explanation and references as composition measurements with EDS is not right technique to confirm tribofilm formation.
Response 15: Thank you for the valuable comments and questions. They were very helpful in improving the quality of the paper. In accordance with your suggestion, we have revised the manuscript to include the following statement: ``Similar to the other three samples, the surface morphology of the reference sample (Ru) after use in the engine operation, shown in Figure 12a, was analyzed. It was observed that, in this case, the residue film is in the form of flakes that cover the entire surface of the valve plate, specific to the initial stages of film formation according to R. Flo el al. [64,65]. This aspect is highlighted in Figure 12 b, where a detail of the residue film flake is presented.`` (lines 436-441)
Comments 16: Authors mentions that ‘However, it is observed that, in the case of this sample, the deposited tribofilm has the smallest thickness, being in fact in the form of flakes arranged on the entire surface of the valve plate.’ Please measure the thickness of tribofilm to validate this claim, as generally the tribfilms are hardly few nm thick.
Response 16: Thank you for the valuable comments and questions. They were very helpful in improving the quality of the paper. In accordance with your suggestion, we have revised the manuscript to include the following statement: ``It was observed that, in this case, the residue film is in the form of flakes that cover the entire surface of the valve plate, specific to the initial stages of film formation according to R. Flo el al. [64,65]. This aspect is highlighted in Figure 12 b, where a detail of the residue film flake is presented.`` (lines 437-441)
Comments 17: Please modify table 7 by replacing all values containing ‘,’ with ‘.’.
Response 17: Thank you for your advice on improving the wording of the paper. Following your suggestion, I have revised the table 7 by replacing all values containing ‘,’ with ‘.’.
Comments 18: Please provide the JCPDS reference number for ‘Figure 12. XRD patterns of the three types of TBC considered: a) Cr3C2 - Ni20Cr; b) MgZrO3 - 35NiCr; c) ZrO2 - 5CaO (with an Al2O3- 30 (Ni20Al) BC in all the cases)’ for all compounds
Response 18: Thank you for your advice, which has been instrumental in improving the scientific quality of the manuscript. Following your guidance, as you will be able to see in the paper, we have added for each of the analyzed compounds the reference code of the database we have used for their identification.
Comments 19: Authors mentions that ‘In contrast to the three samples previously analyzed, it should be emphasized that in the case of the control sample, the presence of the carbon element is observed in a much smaller percentage, which confirms the appearance of the thin, uneven tribofilm distributed on the surface of the valve’ I again request to confirm the presence of tribfilms and their thickness with XPS as it is the right technique to identify the formation of tribofilms even when the thickness of such films is less than 10nm and not XRD, as XRD is bulk characterization technique used for phase identification.
Response 19: Thank you for the valuable comments and questions. They were very helpful in improving the quality of the paper. Following your observation, we removed completely this statement.
Comments 20: Authors mentions ‘the diffractograms obtained being comparatively presented in Figure 13.’, please correct it, as it should be figure 12.
Response 20: Thank you for your advice on improving the wording of the paper. Following your suggestion, we have modified the figure number.
Comments 21: Please modify the conclusion with the actual experimental values, the coatings composition/architecture, why they are better compared to other materials used in this experiment rather than providing the generalized statements of the observed trend.
Response 21: Thank you for your valuable observation, which guided us in adding explanations where our previous descriptions were lacking. In accordance with your suggestion, we have completely revised the Conclusion chapter.
- Response to Comments on the Quality of English Language
Point 1: The English is fine and does not require any improvement.
Response 1: We appreciate your valuable feedback, and we have meticulously reviewed the entire manuscript to guarantee both grammatical precision and academic rigor. Your comments have been instrumental in enhancing the quality of the paper, and we are confident that the revised version now meets the grammar and academic requirements.

Round 2
Reviewer 2 Report
Comments and Suggestions for Authors
Thank you for your revision. I don't agree with XRD part. Please remove the indexing of minor intensity XRD peaks and replace. Fig 13a P1U - index only high intensity 5 peaks and do not index the lower intensity peaks. It doesn't make any sense. and Fig 13c - P3U i didn't see any peaks. the author may repeat the measurements with proper sample height alignment.
Figure 5,6,7 and 9 - EDS images quality is not clear- better remove it, because table information on EDS values are sufficient
Author Response
Comments 1: I don't agree with XRD part. Please remove the indexing of minor intensity XRD peaks and replace. Fig 13a P1U - index only high intensity 5 peaks and do not index the lower intensity peaks. It doesn't make any sense. and Fig 13c - P3U i didn't see any peaks. the author may repeat the measurements with proper sample height alignment.
Response 1: Thank you for the valuable comments and questions. They were very helpful in improving the quality of the paper and we hope that the answer is according to your expectations. In the case of sample P3U we repeated the measurements with the correction of the scan settings, re-interpreted the peaks and indexed the main peaks, as you can see in figure 13c P3u. We made indexing changes also in the case of figure 13a P1u according to your observations.
Comments 2: Figure 5,6,7 and 9 - EDS images quality is not clear- better remove it, because table information on EDS values are sufficient
Response 2: Thank you for your valuable advice, which guided us in adding explanations where our previous descriptions were lacking. Following your suggestion, we have revised the manuscript to include the EDS images at a better quality, as you can see in figures 5,6,7 and 9 of the second corrected version. Their presence in the first revised version of the manuscript was requested by Reviewer 3.

Reviewer 3 Report
Comments and Suggestions for Authors
The authors have incorporated most of the reviewer's suggestion, that has improved the quality of the manuscript, and can be accepted in the present form.
Author Response
Summary:
``The authors have incorporated most of the reviewer's suggestion, that has improved the quality of the manuscript, and can be accepted in the present form.``
Response: Estimeed Reviewer, we would like to express our heartfelt gratitude for the time and effort you have dedicated to reviewing our paper and providing us with your valuable comments and suggestions. We are sincerely grateful for your thoughtful contribution and support throughout the review process.
